# IACW: Intent-Aware Controllable Watermarking for Scalable Authorial Intent Attribution

Hao Huang [1 2]   Ruihua Zhou [1 2]   Jiatang Luo [3 4]   Yunpeng Li [2]   Yuling Liu [1 2]

## Abstract

As Large Language Models (LLMs) integrate into writing workflows, precise governance requires distinguishing "how AI participated" rather than merely "whether AI was used." Traditional binary detection often misclassifies "AI-polished" content as generated, creating fairness risks. We propose shifting from passive post-hoc detection to active intent attribution, focusing on the distinction between Editing (source-anchored) and Generation (unanchored). We introduce **IACW-Instruct**, a corpus of diverse editing operations constructed via a Director–Actor–Judge pipeline to enable systematic evaluation. Building on this benchmark, we propose **Intent-Aware Controllable Watermarking (IACW)**, featuring intent-adaptive entropy gating for semantically lossless embedding. Experiments show that IACW achieves 95% attribution accuracy under 20% token deletion while preserving near-unwatermarked semantic fidelity, establishing a practical paradigm for fine-grained provenance.

## 1. Introduction

Large language models have become integral to daily writing, from drafting emails to polishing academic manuscripts (Lin, 2025; Chakrabarty et al., 2024; Mysore et al., 2025). The critical governance question is no longer "whether AI was used" but "how AI participated." Specifically, did the user intend to **edit** existing content, or to **generate** new content from scratch? We term this distinction **Intent Attribution**: Editing versus Generation (Figure 1).

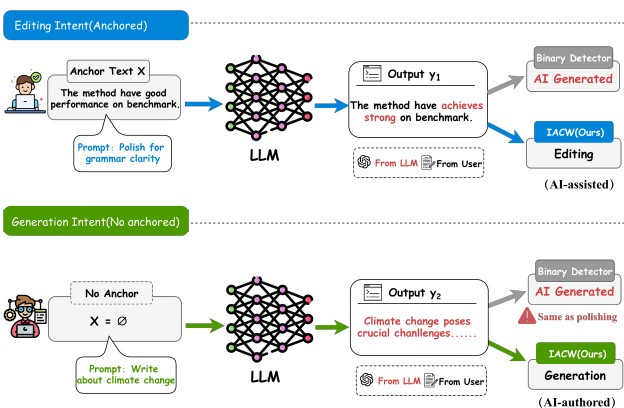

*Figure 1.* The paradigm shift from post-hoc detection to proactive intent attribution via purpose-bound watermarking.

Traditional binary detection ("AI or not?") fundamentally misframes the governance question (Pudasaini et al., 2024; Zhong et al., 2024): it cannot distinguish legitimate editing assistance from wholesale generation, and its elevated false-positive rates disproportionately penalize non-native speakers who rely on AI for legitimate polishing (Liang et al., 2023). More fundamentally, as LLM output distributions converge toward human writing, reliable post-hoc detection faces fundamental statistical limits (Sadasivan et al., 2025).

A promising alternative is **proactive watermarking**: embedding intent signals during generation rather than detecting them post-hoc. However, existing multi-bit watermarking methods apply uniform injection strength across all task types (Yoo et al., 2024), which in many cases compromises the original task objectives for the sake of watermark embedding, a problem especially acute for **polishing tasks**. This uniform approach creates a fundamental tension between **fidelity and robustness**: in polishing scenarios where semantic alteration is nearly impermissible, the injection strength required for decodability induces unacceptable semantic drift (Pang et al., 2024; Fu et al., 2024). Moreover, no dataset offers intent annotations for development.

**Intent-Aware Controllable Watermarking (IACW)** addresses both data and method gaps. For data, we construct **IACW-Instruct**, a 5,000-instance corpus covering diverse editing operations and generation tasks, built via

---
[1]Institute of Information Engineering, Chinese Academy of Sciences, Beijing, China [2]School of Cyber Security, University of Chinese Academy of Sciences, Beijing, China [3]School of Advanced Interdisciplinary Sciences, University of Chinese Academy of Sciences, Beijing, China [4]Institute of Computing Technology, Chinese Academy of Sciences, Beijing, China. Correspondence to: Yunpeng Li <liyunpeng@iie.ac.cn>.

*Proceedings of the 43rd International Conference on Machine Learning*, Seoul, South Korea. PMLR 306, 2026. Copyright 2026 by the author(s).

a **Director–Actor–Judge** pipeline grounded in real user profiles. For method, we propose **Entropy-Gated Embedding** with intent-adaptive thresholds: strict gating for editing tasks preserves semantic fidelity by embedding only at high-entropy positions (where multiple tokens are equally plausible) and skipping low-entropy positions (where deviation would cause obvious artifacts), while relaxed gating for generation maximizes payload density. At decode time, a **confidence-based erasure decoder** autonomously identifies unreliable segments (including those skipped during embedding) via voting patterns, enabling robust ECC recovery without requiring the position mask recorded at generation time. We encode binary intents (Editing vs. Generation) as maximally separated raw payloads (all-zeros vs. all-ones) before ECC protection to ensure reliable attribution under adversarial perturbations.

Our main contributions are as follows:

- We formalize *Intent Attribution* as a watermarking task, framing the Editing versus Generation distinction as a governance objective beyond binary AI detection.

- **IACW-Instruct**, a 5,000-instance corpus constructed via a Director–Actor–Judge pipeline, provides systematic coverage of the editing/generation intent spectrum for training and evaluation.

- The proposed **IACW** framework combines multi-bit payload encoding with **Intent-Adaptive Entropy Gating**, achieving semantically lossless embedding under polishing constraints.

- Empirical results confirm robust intent recovery (95% under 20% token deletion), payload integrity, and near-unwatermarked fidelity across editing scenarios.

## 2. Problem Setup

### 2.1. Intent Attribution Task

**Definition 2.1** (Intent Attribution Task). Given only the output text $y$, verify the intent label $c \in \{\texttt{Edit}, \texttt{Gen}\}$ that was bound at generation time, without access to the input anchor text $x$ or the generation prompt. Formally, construct a decoder $\mathcal{D} : \mathcal{Y} \to \mathcal{C}$ such that $\mathcal{D}(y) = c$ with high probability. This formulation targets *verifiable provenance* rather than post-hoc inference of latent intent.

We ground this distinction in the concept of **Anchor**: an **Anchored Request** takes a given text as input to be preserved, modified, or transformed, while an **Unanchored Request** creates content from scratch or from only a topic hint. Crucially, we distinguish between an **anchor** $x$ (text to be preserved or transformed) and a **reference** $r$ (background material for context). A user who supplies a full paper but

requests "write a review" provides $r \neq \varnothing$ but $x = \varnothing$, the paper serves as a reference context, not as content to be transformed. This constitutes *document-conditioned generation* and is labeled Generation. We map anchored requests $(x \neq \varnothing)$ to the **Editing** label and unanchored requests $(x = \varnothing)$ to **Generation**; this binary mapping captures the fundamental provenance difference, as Editing outputs blend human and AI contributions while Generation outputs are predominantly AI-authored. While finer-grained taxonomies are possible (e.g., distinguishing proofreading from summarization), we argue that this binary distinction captures the most salient provenance boundary for governance purposes; we further justify this choice from a robustness perspective in Section 4.2.

### 2.2. Threat Model

We assume a **white-box** adversary who knows the watermarking algorithm but *not* the secret key $\kappa$. We focus on two representative attack surfaces: *paraphrasing* and *back-translation*, which are the most practical perturbations that preserve semantic content. The primary adversary goal is *Evasion* (corrupting the watermark to avoid detection). We exclude oracle attacks where content is completely rewritten, as such attacks destroy the semantic carrier common to current text watermarking approaches.

## 3. IACW-Instruct: A Benchmark for Intent Attribution

### 3.1. Benchmark Overview: Instance Schema

**Scale and Coverage.** IACW-Instruct comprises 5,000 instances: 69.5% Editing (3,475 instances) and 30.5% Generation (1,525 instances), with 9 editing sub-types and 10 generation sub-types (Table 1) to ensure comprehensive coverage of real-world copilot interactions. The imbalanced distribution reflects actual copilot usage patterns, where editing is more prevalent than generation.

**Structured Instance Schema.** Each instance carries structured, auditable annotations: a prompt paired with its intent label (Edit/Gen) and subtype, contextual metadata (persona, interaction style, domain); for editing intents, it includes the anchor text $x$ that the user expects the model to transform; for document-conditioned generation, it includes context $r$.

### 3.2. Construction Pipeline: Director–Actor–Judge

We construct IACW-Instruct through a hybrid approach combining real user seeds with controlled synthetic expansion, following a **Director–Actor–Judge (DAJ)** pipeline.

**Seed Collection.** We collect 228 real prompts from 47 participants across diverse backgrounds to ensure authentic

| Intent | Sub-type | Discriminative Keywords |
|---|---|---|
| **Editing** | Polish | fluency, clarity, readability |
| | Tone/Style | formal, casual, professional |
| | Summarize | condense, key points, brief |
| | Rewrite | rephrase, different words |
| | Format | structure, headings, bullets |
| | Translate | *[target language]*, convert |
| | Expand | elaborate, more detail |
| | Grammar | grammatical, errors, fix |
| | Proofread | typos, spelling, punctuation |
| **Generation** | Answer | question, why, how |
| | Write Email | correspondence, message |
| | Write Article | essay, blog post, piece |
| | Analyze | review, critique, discuss |
| | Explain | concept, what is |
| | Write Code | implement, function |
| | Write Story | narrative, fiction |
| | Brainstorm | ideas, suggestions |
| | Write Copy | marketing, advertisement, ad |
| | Outline | structure, framework |

*Table 1.* IACW-Instruct intent taxonomy. Editing sub-types require anchor $x$ (content to transform); Generation sub-types have $x = \varnothing$ (may include reference $r$ for context). Full distribution in Appendix B.

intent expressions and natural language variation.

**DAJ Pipeline.** Starting from these seeds, the **Director** generates diverse user profiles that vary in persona, domain, and interaction style, then derives natural prompt variants that preserve the original intent. These prompts are passed to the **Actor**, which produces outputs and, for editing tasks, supplies anchor text $x$ sampled from Wikipedia with controlled corruption (typos, grammar errors) to simulate realistic polishing scenarios. A **Judge** module then filters the resulting instances, retaining only those that exhibit consistent intent, well-aligned outputs, and natural phrasing. We apply strict filtering throughout; details appear in Appendix A. All components use GPT-4o (OpenAI, 2024) with temperature 0.9. Director generates ∼29 candidate variants per seed; after Judge filtering (76.6% acceptance rate), each seed yields ∼22 retained instances, totaling 5,000. Participants provided informed consent for research use, and all collected prompts were anonymized to remove personally identifiable information before use.

**Intent Labeling Protocol.** Building on the anchor/reference distinction (Section 2.1), we operationalize labeling through explicit boundary rules. The key discriminator is whether the instruction imposes a *preservation constraint* requiring transformation of specific content (anchor $x$) versus merely using material for context (reference $r$): presence of anchor implies Editing, absence implies Generation. Three edge cases require special handling: (1) **Summarization** is labeled Editing when

condensing anchor content without adding claims, but Generation when permitting interpretive synthesis (e.g., "write an executive summary with recommendations"); (2) **Expansion** is labeled Editing when the instruction requires preserving original content while adding depth (e.g., "elaborate on each point"), but Generation when the draft serves only as a topic seed (e.g., "write more about this topic"); (3) **Document-conditioned tasks** (review, critique) are labeled Generation because the input serves as reference $r$, not anchor $x$; they produce new content *about* the document rather than transforming it. Ambiguous cases blending polishing with elaboration are rejected. Quality control statistics are reported in Appendix A.

## 4. Intent-Aware Controllable Watermarking

We present Intent-Aware Controllable Watermarking (IACW), a framework that binds a generation-time intent signal into a recoverable watermark payload. The pipeline has three stages: payload encoding, selective entropy-gated embedding, and erasure-aware decoding. Intent binding maps a provided intent label to a payload, entropy gating reduces distortion under polishing constraints, and confidence-based decoding exploits erasure structure for robust recovery without requiring generation-time side information.

### 4.1. Multi-bit Backbone

We instantiate the pseudo-random segment-assignment backbone of Qu et al. (Qu et al., 2025). This subsection restates the inherited backbone in self-contained notation; our modifications start from the intent payload, the entropy gate, and the erasure-aware decoder.

Let $\mathcal{V}$ be the vocabulary, $q$ the prompt or application context, and $y = (y_1, \ldots, y_T)$ the generated token sequence, with $y_0$ denoting the beginning-of-sequence token. Let $\kappa$ be the secret key, $\ell_t(v)$ the pre-watermark logit of candidate token $v \in \mathcal{V}$ at generation step $t$, and $\delta$ the watermark bias strength. Given a payload $\mathbf{u}$, ECC encoding produces an encoded segment sequence

$$E = \text{Enc}_{\text{ECC}}(\mathbf{u}) = (E[1], \ldots, E[n]), \quad (1)$$

where $n$ is the number of watermark segments and $E[p]$ is the value assigned to segment $p$. In our BCH instantiation, segment values are binary; the notation follows the same segment-value interface as Qu et al.

Following Qu et al., we construct a secret-keyed token-to-segment mapping

$$M_\kappa : \mathcal{V} \to \{1, \ldots, n\}, \quad (2)$$

optimized for balanced segment allocation under natural token frequencies. At generation step $t$, the previous token,

rather than the step index itself, selects the segment:

$$p_t = M_\kappa(y_{t-1}). \tag{3}$$

Thus $p_t$ is the selected segment index, while $E[p_t]$ is the segment value used at step $t$.

The green list is then determined by a keyed pseudo-random function conditioned on the secret key, previous token, and current segment value:

$$G_t(E[p_t]) = \{v \in \mathcal{V} : h(\kappa, y_{t-1}, E[p_t], v) < \rho\}, \tag{4}$$

where $h$ is normalized to $[0, 1]$ and $\rho = 0.5$ gives an equal green/red partition. The backbone biases logits toward the green list:

$$\ell'_t(v) = \ell_t(v) + \delta \cdot \mathbb{I}[v \in G_t(E[p_t])]. \tag{5}$$

At extraction time, the decoder reconstructs segment votes using only the text and the secret key. For each token $y_t$, it recomputes $p_t = M_\kappa(y_{t-1})$; for each candidate segment value $a$, it recomputes the corresponding green list and increments $\mathrm{COUNT}[p_t, a]$ when $y_t$ belongs to that list. The backbone segment estimate is

$$\hat{E}[p] = \arg\max_a \mathrm{COUNT}[p, a], \tag{6}$$

after which ECC decoding recovers the payload:

$$\hat{\mathbf{u}} = \mathrm{Dec}_{\mathrm{ECC}}(\hat{E}). \tag{7}$$

**Our Extensions.** The segment-assignment and green-list construction above are inherited from Qu et al.; IACW does not claim these components as new. We extend the backbone along two dimensions (Figure 2):

- **Intent-Adaptive Entropy Gating** (Section 4.3): We selectively embed at high-entropy positions and skip low-entropy positions prone to artifacts, enabling semantically lossless watermarking for editing tasks.
- **Confidence-Based Erasure Decoding** (Section 4.4): We identify unreliable segments at decode time via voting confidence, treating them as erasures rather than forced errors for erasure-aware ECC recovery.

### 4.2. Intent Binding

Before embedding, we encode the bound intent signal into a payload designed for maximum separability.

**Payload Encoding.** The intent signal is encoded through a deterministic pipeline:

$$c \xrightarrow{\phi} \mathbf{u} \xrightarrow{\mathrm{ECC}} E \tag{8}$$

where $\phi : \mathcal{C} \to \{0, 1\}^d$ maps intent category $c$ to a raw $d$-bit payload, and ECC encoding produces the segment sequence $E$ used by the multi-bit backbone.

**Maximum Separation Design.** We adopt binary attribution rather than fine-grained categories, as finer classification reduces inter-class Hamming distance and thus adversarial tolerance. For binary attribution (Editing vs. Generation), we employ a **maximum separation** strategy:

$$\phi(\texttt{Edit}) = \mathbf{0}^d \quad \text{(all-zeros)} \tag{9}$$
$$\phi(\texttt{Gen}) = \mathbf{1}^d \quad \text{(all-ones)} \tag{10}$$

This maximizes the Hamming distance between the two raw payloads. ECC is not used to exceed the $d/2$ limit of majority voting under pure bit-flip corruption; rather, it improves recovery when entropy gating, deletion, paraphrasing, or weak segment evidence creates erasures or clustered segment errors.

**Intent Source Abstraction.** The intent label $c$ bound into the watermark can originate from multiple sources: (i) **explicit declaration** from the user interface or API call, (ii) **policy-based assignment** where deployment context determines intent (e.g., a "polishing endpoint" always binds Editing), or (iii) **classifier inference** as a fallback when explicit signals are unavailable. We denote this abstraction as $c \leftarrow \texttt{IntentSource}(\text{context})$. Our contribution is the robust binding and verification of $c$, orthogonal to how $c$ is determined; the classifier is one instantiation of IntentSource, not the security root.

**Intent Detection at Deployment.** At generation time, an intent classifier (LLM zero-shot or fine-tuned cross-encoder) serves as one possible IntentSource, inferring intent from the prompt and selecting the corresponding payload. Crucially, **verification requires only the watermarked text**; no classifier access is needed at attribution time. The generation-side IntentSource determines which payload to embed; once embedded, the recovered payload alone determines attribution. This decouples deployment (where IntentSource runs) from verification (where only text is available). When the IntentSource confidence is low or decoding yields ambiguous results (the recovered one-bit fraction is near 0.5), the system outputs `Unknown` rather than forcing a potentially incorrect attribution, prioritizing precision over recall.

### 4.3. Entropy-Gated Embedding

The embedding phase selectively injects the watermark, skipping positions where biasing would corrupt intended semantics.

**Token Entropy.** At each generation step $t$, let $\pi_t(v)$ denote the probability assigned to token $v \in \mathcal{V}$ by the LLM. We define the **semantic entropy** as:

$$H_t = -\sum_{v \in \mathcal{V}} \pi_t(v) \log \pi_t(v) \tag{11}$$

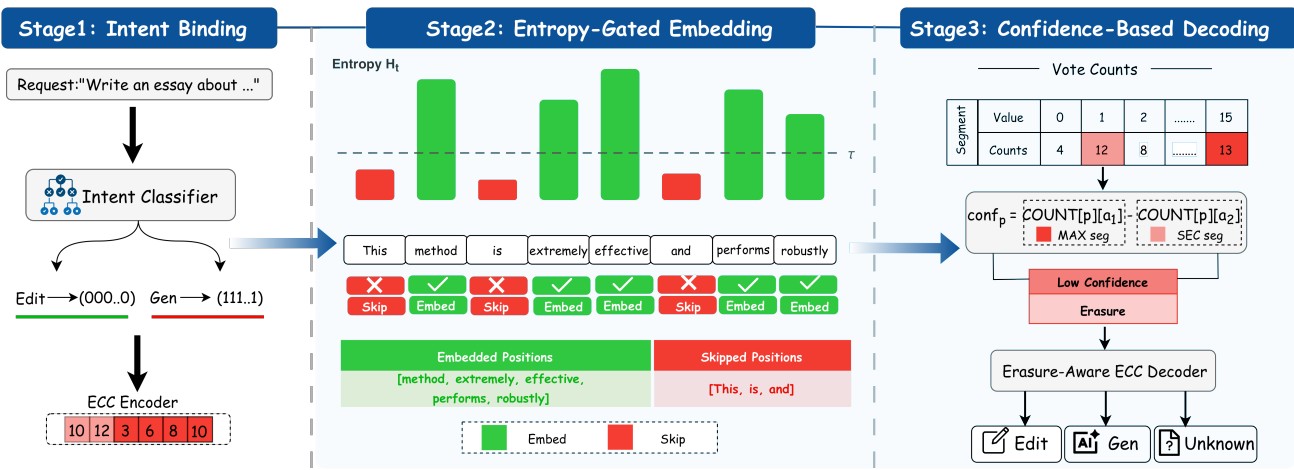

*Figure 2.* IACW pipeline overview. **Left**: Intent binding encodes the intent label into an ECC-protected segment sequence. **Center**: Entropy gating selectively embeds watermark bits at high-entropy positions (green) where multiple tokens are equally plausible, while skipping low-entropy positions (red) where biasing would cause artifacts. **Right**: The decoder independently identifies unreliable segments via voting confidence and treats them as erasures, enabling robust recovery without generation-time side information.

High $H_t$ indicates multiple semantically plausible continuations; biasing at such positions introduces minimal distortion since no single token dominates. Conversely, low-entropy positions represent near-deterministic completions where biasing is more likely to produce noticeable artifacts.

**Gating Decision.** The injection gate at step $t$ is:

$$g_t = \mathbb{I}[H_t \geq \tau] \tag{12}$$

where $\tau$ is an intent-adaptive threshold. Combining with Eq. (5), the modified logits become:

$$\ell'_t(v) = \ell_t(v) + g_t \cdot \delta \cdot \mathbb{I}[v \in G_t(E[p_t])] \tag{13}$$

When $g_t = 1$ (high entropy, safe to embed), we apply full bias $\delta$ toward the green list; when $g_t = 0$ (low entropy, risk of artifacts), we skip injection entirely and sample from the unmodified distribution. Crucially, the segment assignment $p_t = M_\kappa(y_{t-1})$ (Eq. (3)) still applies; the position contributes to decoding votes but without watermark bias.

**Intuition.** High-entropy positions correspond to semantically flexible decision points where multiple tokens are equally plausible; biasing toward green tokens here introduces minimal perceptual distortion. Low-entropy positions represent near-deterministic continuations (e.g., completing "New York" after "New"), where any deviation would produce obvious artifacts. By skipping these rigid positions, IACW preserves semantic fidelity while maintaining sufficient injection density for payload recovery.

**Intent-Adaptive Threshold.** The threshold $\tau$ is set according to detected intent:

---

**Algorithm 1** IACW Embedding

---

**Require:** Prompt/context $q$, intent $c$, key $\kappa$, bias $\delta$
**Ensure:** Watermarked $y$
1: $\mathbf{u} \leftarrow \phi(c)$            *// raw intent payload*
2: $E \leftarrow \text{Enc}_{\text{ECC}}(\mathbf{u})$     *// encode intent to $n$-segment sequence*
3: $\tau \leftarrow \tau_c$           *// intent-adaptive threshold*
4: **for** $t = 1, \ldots, T$ **do**
5:     $\ell_t \leftarrow \text{LM}(q, y_{<t})$     *// get logits from LLM*
6:     $p_t \leftarrow M_\kappa(y_{t-1})$     *// segment index (Eq. (3))*
7:     $G_t \leftarrow \{v \in \mathcal{V} : h(\kappa, y_{t-1}, E[p_t], v) < \rho\}$   *// Eq. (4)*
8:     $H_t \leftarrow \text{Entropy}(\text{softmax}(\ell_t))$
9:     **if** $H_t \geq \tau$ **then**
10:       $\ell'_t(v) \leftarrow \ell_t(v) + \delta \cdot \mathbb{I}[v \in G_t]$   *// high entropy: apply bias*
11:       $y_t \sim \text{softmax}(\ell'_t)$     *// sample from biased distribution*
12:     **else**
13:       $y_t \sim \text{softmax}(\ell_t)$     *// low entropy: skip bias*
14:     **end if**
15: **end for**
16: **return** $y$

---

- **Editing** ($\tau_{\text{edit}} = 0.5$ nats): Strict gating to preserve author semantics.
- **Generation** ($\tau_{\text{gen}} = 0.3$ nats): Relaxed gating to maximize payload density.

This asymmetry reflects the differential fidelity requirements: polishing tasks typically require semantically lossless transformation, while generation tasks tolerate more aggressive embedding. Threshold values are selected via grid search on the development set.

### 4.4. Confidence-Based Erasure Decoding

A key technical contribution is our decoding strategy: rather than requiring the erasure mask from generation, the decoder **independently identifies unreliable segments** via voting confidence. This makes the decoder fully autonomous: it operates using only the text and secret key.

**Vote Counting.** For each token in the text, the decoder reconstructs the green/red list assignments via context hashing and checks membership. Votes accumulate in a matrix COUNT$[p, a]$, where $p$ indexes segments and $a$ indexes candidate segment values.

**Confidence-Based Erasure Detection.** For each segment $p$, we compute the vote margin between the top-1 and top-2 candidates:

$$\text{conf}_p = \text{COUNT}[p, a_1] - \text{COUNT}[p, a_2] \qquad (14)$$

Segments with $\text{conf}_p < \theta$ are flagged as *erasures* rather than forced decisions. The primary motivation is **robustness against adversarial attacks**: when tokens are deleted or paraphrased, the affected positions contribute noisy votes, reducing the confidence margin of corresponding segments. By marking these low-confidence segments as erasures rather than forcing potentially incorrect decisions, the decoder leverages the ECC's erasure-correction capability. Crucially, the same mechanism also handles entropy gating gracefully: tokens at skipped positions (where $g_t = 0$) produce near-uniform votes that similarly reduce segment confidence; thus, a single unified framework addresses both attack robustness and quality-preserving skips, *without requiring explicit communication of the skip mask or attack detection*.

**Erasure-Aware ECC Decoding.** Modeling uncertain segments as erasures rather than forcing potentially incorrect decisions provides two benefits: (1) when the number of unreliable segments exceeds the error-correction capacity $r$ but remains within $2r$, erasure-aware decoding can still recover the payload, since a code correcting $r$ errors can correct $2r$ erasures; (2) when erasures exceed $2r$, the system gracefully fails and classifies the text as *unwatermarked*, either human-authored or generated by a non-participating LLM, avoiding incorrect attribution. The decoder passes segment values and erasure flags to the ECC decoder, which recovers the original payload $\hat{\mathbf{u}}$.

**Margin-Based Attribution.** To ensure robust decisions when evidence is weak, we introduce a symmetric margin $\gamma$ around the neutral point 0.5. Let $\bar{u} = d^{-1} \sum_i \hat{u}_i$ denote the proportion of ones in the recovered payload:

$$\hat{c} = \begin{cases} \texttt{Gen} & \text{if } \bar{u} \geq 0.5 + \gamma \\ \texttt{Edit} & \text{if } \bar{u} \leq 0.5 - \gamma \\ \texttt{Unknown} & \text{otherwise} \end{cases} \qquad (15)$$

The margin $\gamma$ defines an abstention zone $[0.5 - \gamma, 0.5 + \gamma]$: when the recovered signal falls within this interval, the evidence is insufficient for confident attribution. We set $\gamma$ proportional to the ECC's normalized correction capacity, ensuring that the abstention threshold reflects the code's inherent uncertainty bound; if the fraction of potentially corrupted bits approaches the correction limit, the decoder abstains rather than risking misattribution. This fail-safe design ensures graceful degradation: under adversarial edits or signal loss, the system defaults to `Unknown` rather than false accusation.

## 5. Experiments

We evaluate IACW along three axes: (RQ1) *Semantic fidelity*, asking whether entropy gating enables "semantically lossless" embedding; (RQ2) *Robustness* under adversarial attacks such as paraphrasing and back-translation; and (RQ3) *Realistic editing workflow*, where watermarked content is diluted by human-written text in mixed documents.

### 5.1. Experimental Setup

**Datasets.** We evaluate on the held-out test set of IACW-Instruct (200 instances, including editing and generation). For **Editing** tasks, we use Wikipedia articles as anchor texts across polishing, rewriting, summarization, and formatting. For **Generation** tasks, we include article writing, email composition, and creative generation.

**Implementation.** We use Llama-3-8B-Instruct (Meta AI, 2024) as the base model with nucleus sampling ($p$=0.95) modified by watermark bias ($\delta = 2.5$, greenlist ratio $\rho = 0.5$). Entropy thresholds are set adaptively: $\tau_{\text{edit}} = 0.5$ nats for editing, $\tau_{\text{gen}} = 0.3$ nats for generation. The intent payload uses a $(15, 7)$ BCH code with 2-bit error correction. All experiments run on $4\times$ NVIDIA A100 GPUs.

**Baselines.**

- **KGW** (Kirchenbauer et al., 2023): The seminal zero-bit watermarking method that partitions vocabulary via context-dependent hashing and biases sampling toward green tokens.
- **Qu et al.** (Qu et al., 2025): Multi-bit watermarking with constant bias at all positions (no adaptive gating). Serves as our backbone.
- **Random Skip**: Same backbone as Qu et al., but skips the same proportion of positions as IACW with random

*Table 2.* Watermark detectability on clean text. KGW, Qu et al., and IACW achieve perfect detection.

| Method | TPR | FPR | AUC |
|---|---|---|---|
| *Zero-bit Watermarking* | | | |
| KGW (Kirchenbauer et al., 2023) | 1.000 | <0.05 | 1.000 |
| *Multi-bit Watermarking* | | | |
| Qu et al. (Qu et al., 2025) | 1.000 | <0.05 | 1.000 |
| Random Skip | 0.000 | <0.05 | 0.500 |
| **IACW (Ours)** | 1.000 | <0.05 | 1.000 |

selection rather than entropy-based.

**Metrics.** We report **TPR** (true positive rate on watermarked text), **FPR** (false positive rate on unwatermarked text), **AUC** (area under the ROC curve), **Intent Accuracy** (correct Edit/Gen attribution), **Edit Ratio** (Levenshtein distance / source length), and **PPL** (perplexity via GPT-2 Large (Radford et al., 2019)).

## 5.2. Watermark Detectability

Table 2 shows that KGW, Qu et al., and IACW all achieve 100% TPR at <0.05 FPR, confirming sufficient watermark strength for downstream attribution. In contrast, Random Skip fails completely (0% TPR), demonstrating that arbitrary position skipping destroys the watermark signal— entropy-guided selection is essential. Both zero-bit (KGW) and multi-bit methods (Qu et al., IACW) achieve perfect detection on clean text, showing that entropy-gated skipping does not compromise detectability when texts are unperturbed. Standard single-key zero-bit schemes such as KGW answer whether a text is watermarked under a given key, but do not carry an explicit payload. A binary attribution variant can be constructed by assigning separate keys to Editing and Generation and selecting the stronger detector response. However, this key-enumeration approach scales linearly with the number of intents and introduces a coverage–false-attribution trade-off under limited token budgets. We therefore use multi-bit payload encoding with erasure-aware decoding, and compare against a KGW-Dual-Key baseline in Appendix D. For intent attribution, IACW achieves 99.5% accuracy in correctly distinguishing editing from generation intent based solely on output text.

## 5.3. Semantic Fidelity

For editing tasks, users expect near-lossless transformation. We evaluate whether entropy gating preserves source semantics while maintaining attribution capability.

**Results.** Figure 3 shows that IACW approaches unwatermarked quality across all metrics, while Qu et al. degrades significantly. The key insight: *which positions to skip mat-*

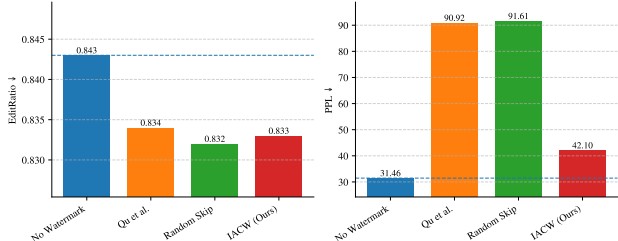

*Figure 3.* Semantic fidelity on editing tasks. IACW (red) approaches unwatermarked quality (blue baseline) across all metrics, while Qu et al. (orange) degrades significantly. Dashed lines indicate the No Watermark reference.

*ters.* Random Skip, despite skipping the same proportion of tokens, fails to match IACW's fidelity because it does not target semantically critical positions. Entropy-guided gating identifies low-entropy positions where biasing would cause obvious artifacts, skipping them to preserve author semantics while embedding at high-entropy positions to maintain sufficient injection density for payload recovery.

**Human Evaluation.** We validate low-distortion behavior via human study (3 annotators, 150 samples). IACW achieves 48.2% preference rate vs. No Watermark (near-chance), compared to 38.7% for Qu et al. Annotators detected watermarks in only 52.2% of IACW outputs (near-random), versus 67.3% for Qu et al. Table 3 provides a qualitative example demonstrating that watermarked outputs maintain the requested professional tone while preserving semantic equivalence.

## 5.4. Robustness

We evaluate attribution accuracy under paraphrasing, back-translation, and token-level perturbations.

**Results.** Table 4 shows that IACW outperforms Qu et al. across all attack scenarios. Under 20% token deletion, IACW maintains 95.0% intent accuracy compared to Qu et al.'s 90.5%. Paraphrasing and back-translation pose greater challenges, reducing accuracy to 69.0% and 54.5% respectively, though IACW still substantially outperforms Qu et al. (58.0% and 36.5%). Both methods degrade gracefully under increasing perturbation.

## 5.5. Human-AI Mixed Workflow

**Experimental Design.** To evaluate IACW under realistic mixed-authorship scenarios, we construct documents by mixing human-written paragraphs with AI-polished (watermarked) paragraphs at varying ratios. Specifically, we construct documents with AI-polished content ratios $\alpha \in \{10\%, 20\%, 30\%, 50\%, 70\%, 100\%\}$, where $\alpha = 100\%$ represents fully watermarked output and $\alpha = 10\%$

*Table 3.* Qualitative example of semantic fidelity. The watermarked output preserves the professional tone requested while maintaining semantic equivalence to the reference completion.

| Prompt | Reference | Watermarked Text |
|---|---|---|
| Please polish to a more professional tone for a work report. | Virtual environment software refers to any software or system that implements and manages virtual computing environments ... | Virtual environment software encompasses any application or system designed to implement and maintain virtualized computing infrastructures ... |

*Table 4.* Intent accuracy (%) under various attacks. IACW outperforms Qu et al. across all attack scenarios while preserving superior fidelity.

| Method | Clean | Paraph. | Del-20% | BackTrans |
|---|---|---|---|---|
| Qu et al. | 93.0% | 58.0% | 90.5% | 36.5% |
| **IACW (Ours)** | **99.5%** | **69.0%** | **95.0%** | **54.5%** |

*Table 5.* Detectability and intent attribution under human-AI content mixing. IACW maintains practical utility even when only 30% of the content is AI-polished. At $\alpha = 20\%$, detection remains reliable while attribution begins to degrade gracefully.

| Metric | AI-Polished Ratio $\alpha$ | | | | | |
|---|---|---|---|---|---|---|
| | 10% | 20% | 30% | 50% | 70% | 100% |
| *IACW (Ours)* | | | | | | |
| Detectability (%) | 91.0 | 93.5 | 98.0 | 100.0 | 100.0 | 100.0 |
| Intent Acc. (%) | 64.0 | 76.5 | 83.0 | 94.0 | 97.5 | 99.5 |
| *KGW (Zero-bit baseline)* | | | | | | |
| Detectability (%) | 0.0 | 0.0 | 0.0 | 6.0 | 58.0 | 100.0 |
| Intent Acc. (%) | - | - | - | - | - | - |

represents a document where only one paragraph in ten was AI-polished. Human paragraphs are sampled from Wikipedia and news articles; AI-polished paragraphs use IACW-watermarked editing outputs. We evaluate: (1) **Detectability**: can we identify documents containing *any* watermarked content? (2) **Intent Attribution**: can we correctly recover the Editing intent from diluted signals?

**Results.** Table 5 reveals graceful degradation under content dilution. Detectability remains above 90% even at the lowest tested ratio (91.0% at $\alpha = 10\%$), sufficient for most practical scenarios where users polish at least a few paragraphs. Intent attribution shows slightly steeper degradation but maintains >80% accuracy at $\alpha = 30\%$, the regime where roughly one-third of paragraphs are AI-polished.

Notably, IACW outperforms KGW on detectability at low $\alpha$ values despite KGW's stronger signal density. This counterintuitive result arises because IACW's entropy-gated embedding concentrates the watermark signal in high-entropy positions that are less likely to be paraphrased during human editing, whereas KGW's uniform embedding spreads the signal across positions that may be overwritten when humans revise adjacent paragraphs.

*Table 6.* Ablation study. Entropy gating affects text fidelity, while ECC and erasure decoding affect payload recovery. PPL is reported only for variants that change generated text.

| Configuration | Intent Acc.↑ | PPL↓ |
|---|---|---|
| **IACW (Full)** | **99.5%** | 42.1 |
| w/o Entropy Gating | 99.0% | 89.1 |
| w/o ECC | 90.0% | – |
| w/o Erasure Decoding | 85.5% | – |

**Implications for Governance.** These results inform deployment guidelines: (1) For document-level provenance ("does this contain AI assistance?"), IACW provides reliable detection even with sparse AI involvement ($\alpha \geq 20\%$). (2) For intent attribution, practitioners should expect reduced confidence when AI-polished content comprises <30% of the document. (3) Paragraph-level attribution, i.e., identifying *which* segments are AI-polished, remains an open challenge that we leave for future work.

### 5.6. Ablation Study

**Results.** Table 6 shows complementary roles for generation-side and recovery-side components. Removing entropy gating degrades fidelity, increasing PPL from 42.1 to 89.1, while removing ECC or erasure-aware decoding reduces intent accuracy to 90.0% and 85.5%, respectively. Thus, IACW preserves text quality through entropy gating and improves attribution robustness through ECC-based recovery.

## 6. Related Work

**LLM Watermarking.** Unlike post-hoc detectors that infer generation from output statistics alone (Mitchell et al., 2023; Gehrmann et al., 2019), zero-bit watermarking schemes (Kirchenbauer et al., 2023; Zhao et al., 2023) embed statistical signals at generation time to detect AI-generated text, but still answer only presence, not intent. Multi-bit schemes (Boroujeny et al., 2024; Qu et al., 2025; Yoo et al., 2024) embed informative payloads for finer-grained provenance, yet apply payloads uniformly across positions. CredID (Jiang et al., 2025) and certified-robustness methods (Feng et al., 2024; Zhao et al., 2024) provide provable guarantees but similarly ignore that different intents impose different fidelity constraints.

**Adaptive and Entropy-based Watermarking.** Prior work uses entropy to balance robustness and quality (Wang et al., 2025; Lee et al., 2024; Fu et al., 2024; Pang et al., 2024), skipping low-entropy positions where artifacts are perceptible. However, these methods treat skipping as an empirical heuristic without formal recovery analysis. We instead model skipped positions as erasures within an ECC framework, transforming quality-preserving skips into a tractable channel coding problem.

**Prompt-level Intent & Role Recognition.** Human-AI collaborative writing studies (Mysore et al., 2025) distinguish *anchored* tasks (editing) from *unanchored* tasks (generation). LLM Role Recognition (Cheng et al., 2025) and source-sentence tracing (Zhu et al., 2025) move beyond binary AI detection, but these post-hoc classifiers require prompt access at inference, which is unavailable when only final text is submitted.

## 7. Limitations

**Deployment Constraints.** IACW requires model-level access and cannot watermark outputs from proprietary APIs. Oracle attacks that completely rewrite content can destroy the watermark, a fundamental limitation shared by all text watermarking schemes.

**Attribution Granularity.** Our binary framing trades fine-grained resolution for robustness. Compound requests mixing both intents are labeled by the dominant category; segment-level attribution remains future work.

**Applicability Boundaries.** Texts shorter than 150 tokens may lack sufficient capacity for reliable attribution. Current evaluation focuses on English; multilingual settings require language-aware threshold calibration.

## 8. Conclusion

We introduced Intent-Aware Controllable Watermarking (IACW), a framework that embeds authoring intent into LLM outputs as a recoverable watermark signal. Our key contributions are: (1) intent-adaptive entropy gating that preserves semantic fidelity for editing tasks while maintaining sufficient injection density for generation tasks; (2) confidence-based erasure decoding that enables robust recovery without generation-time side information; and (3) IACW-Instruct, a benchmark of 5,000 instances for systematic evaluation.

Experiments demonstrate that IACW achieves 99.5% attribution accuracy from output text alone, maintains 95.0% accuracy under 20% token deletion, and approaches unwatermarked quality on editing tasks. Future work includes segment-level attribution for mixed-intent documents and extension to multilingual settings.

## Acknowledgments

We thank the anonymous reviewers and the area chair for their constructive feedback. This work is supported by the Beijing High Innovation Plan (No. 202504841069). We used Gemini for polishing the text.

## Impact Statement

This work studies controllable watermarking for attributing authorial intent in generated text. The proposed method may support provenance tracking, responsible disclosure, and accountability for synthetic content. At the same time, watermarking and attribution mechanisms can be misused for surveillance, over-attribution, or unsupported claims of authorship if deployed without transparency and appropriate safeguards. We therefore emphasize that such systems should be used as auxiliary evidence rather than definitive proof, and should be accompanied by clear disclosure, auditing, and appeal mechanisms.

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

## A. Quality Control & Reliability

This appendix reports the quality-control procedures and inter-annotator agreement for constructing IACW-Instruct.

**Rejection Statistics.**    The Judge rejected 23.4% of candidate instances. Table 7 provides the detailed breakdown.

*Table 7.* Judge quality control statistics. Rejection rates by category and human verification agreement.

| Metric | Value |
|---|---|
| **Rejection Breakdown** *(of 23.4% total)* | |
| Intent ambiguity | 41% |
| Output misalignment | 35% |
| Unnatural prompt | 24% |
| **Human Verification** *(n=100)* | |
| Inter-annotator $\kappa$ (Cohen's Kappa) | 0.87 |
| Human–Judge agreement | 91.5% |

**Human Verification Protocol.**    To validate automated judgments, we conducted human verification on 100 randomly sampled instances across all intent categories. Two annotators independently labeled each instance for intent consistency and boundary correctness. Inter-annotator agreement was $\kappa = 0.87$ (Cohen's Kappa), indicating strong agreement. Agreement with automated Judge labels was 91.5%. Disagreements were resolved through discussion and used to iteratively refine Judge prompts.

**Intent Ambiguity Analysis.**    Of the 23.4% total rejection rate, intent ambiguity accounted for 41% of rejections. Common ambiguity patterns include:

- Summarization with creative additions (e.g., "summarize and add your insights")
- Requests blending polishing with open-ended elaboration (e.g., "improve and expand this draft")
- Expansion preservation intent unclear (e.g., "write more about this" without specifying whether to keep original content)

## B. Dataset Statistics & Splits

**Intent Distribution.**    The final dataset contains 5,000 instances: 69.5% Editing (3,475 instances) and 30.5% Generation (1,525 instances). Table 8 shows the complete sub-type distribution.

**Text Length Statistics.**    For editing tasks, the mean anchor text ($x$) length is 287 tokens (std: 124), and the mean output length is 312 tokens (std: 98). For generation tasks, the mean output length is 456 tokens (std: 203).

**Edit Distance Analysis.**    The average edit distance ratio (Levenshtein distance normalized by anchor length) for editing tasks is 0.31, indicating moderate rewriting intensity:

- 34% of editing instances: ratio $< 0.2$ (light polishing)
- 42% of editing instances: ratio 0.2–0.4 (moderate rewriting)
- 24% of editing instances: ratio $> 0.4$ (substantial revision)

This distribution enables evaluation across different fidelity constraint levels.

**Data Splits.**    We partition IACW-Instruct into train/dev/test (81%/15%/4%) using **seed-level grouping**: all variants derived from the same seed prompt are assigned to the same split, preventing near-duplicate leakage across partitions. Splits are further balanced by intent and sub-type distribution:

- **Train**: 4,050 instances from 180 seeds (2,854 Editing, 1,196 Generation)
- **Dev**: 750 instances from 33 seeds (521 Editing, 229 Generation)
- **Test**: 200 instances from 15 seeds (100 Editing, 100 Generation)

*Table 8.* Complete sub-type distribution in IACW-Instruct.

| Editing | | | Generation | | |
|---|---|---|---|---|---|
| **Sub-type** | **Count** | **%** | **Sub-type** | **Count** | **%** |
| Polish | 530 | 10.6 | Answer | 135 | 2.7 |
| Tone/Style | 445 | 8.9 | Write Email | 225 | 4.5 |
| Summarize | 420 | 8.4 | Write Article | 215 | 4.3 |
| Rewrite | 385 | 7.7 | Analyze | 165 | 3.3 |
| Format | 370 | 7.4 | Explain | 140 | 2.8 |
| Translate | 355 | 7.1 | Write Code | 140 | 2.8 |
| Expand | 345 | 6.9 | Write Story | 140 | 2.8 |
| Grammar | 345 | 6.9 | Brainstorm | 135 | 2.7 |
| Proofread | 280 | 5.6 | Write Copy | 125 | 2.5 |
| | | | Outline | 105 | 2.1 |

## C. Human-AI Mixed Workflow Analysis

This appendix details the experimental setup for the human-AI mixed-workflow evaluation in Section 5.5.

**Experimental Setup.** We construct mixed documents by interleaving human-written paragraphs (from Wikipedia, news articles, and academic abstracts) with AI-polished paragraphs (IACW-watermarked editing outputs). Each document contains 8–12 paragraphs totaling 800–1,200 tokens. For each mixing ratio $\alpha \in \{10\%, 20\%, 30\%, 50\%, 70\%, 100\%\}$, we generate 200 documents (100 per intent class), yielding 1,200 evaluation instances.

## D. Additional Experiments

**Multi-model generalization.** We evaluate IACW on two additional instruction-tuned models, Qwen2.5-7B-Instruct and Mistral-7B. The same watermarking pipeline is used without model-specific adaptation, since IACW operates on token probability distributions.

*Table 9.* Intent attribution accuracy across base models. IACW generalizes beyond Llama-3-8B-Instruct without model-specific adaptation.

| Model | Clean Acc. | Del-20% | Paraph. |
|---|---|---|---|
| Llama-3-8B-Instruct | 99.5 | 95.0 | 69.0 |
| Qwen2.5-7B-Instruct | 99.0 | 96.5 | 74.0 |
| Mistral-7B | 93.0 | 88.5 | 73.0 |

**Expanded seed-level evaluation.** The original held-out test split contains 200 instances derived from 15 held-out human seed prompts. To test whether the results depend on this narrow seed set, we additionally evaluate on 100 new human-written seeds, yielding 400 instances.

*Table 10.* Expanded seed-level evaluation. Performance remains strong on a broader 100-seed evaluation set, although shorter outputs reduce the available watermark budget.

| Evaluation Set | Clean Acc. | Del-20% Acc. |
|---|---|---|
| Original held-out split | 99.5 | 95.0 |
| Expanded 100-seed split | 95.0 | 89.5 |

**KGW-Dual-Key baseline.** Although standard single-key KGW detects watermark presence rather than payload identity, a binary attribution baseline can be built by assigning separate keys to Editing and Generation. At detection time, we run both detectors and assign the intent with the stronger response, abstaining when neither exceeds the threshold. This baseline faces a trade-off between attribution coverage and false attribution.

*Table 11.* Comparison with a KGW-Dual-Key attribution baseline. Lowering the z-threshold improves coverage but introduces false attributions, while IACW maintains higher accuracy without false attribution in this setting.

| Method | Threshold | Clean Acc. | False Attr. Rate |
|---|---|---|---|
| KGW-Dual-Key | $z = 3.0$ | 88.0 | 0.0 |
| KGW-Dual-Key | $z = 2.0$ | 93.5 | 5.0 |
| IACW (Ours) | – | 99.5 | 0.0 |

