# OpenReview forum: "IACW: Intent-Aware Controllable Watermarking for Scalable Authorial Intent Attribution"
_ICML.cc/2026/Conference — ICML 2026 regular_

### Official Review · Reviewer_LcbK · 2026-03-12

**Soundness:** 1
**Presentation:** 3
**Significance:** 2
**Originality:** 2
**Overall Recommendation:** 2
**Confidence:** 4

**Summary:**

The paper attempts to watermark text not just to detect if a text is generated by AI but also to discriminate if the AI generated text is just rephrased or if it is generated from scratch. The authors propose to classify the intent (rephrasing vs. from scratch generation) and watermarking the text using two different messages (one for the rephrasing and different one for the from scratch generation). However, the authors do not solve the intent clarification, they only propose how to watermark text using two messages(=keys). Additional contribution of the paper is a dataset of 5000 AI generated prompts with context based on 228 human prompts.

**Compliance With Llm Reviewing Policy:**

Affirmed.

**Final Justification:**

Unfortunately, the author do not address **[W1]** that their evaluation is based only on 15 human prompts. **I simply cannot recommend to accept a paper to ICML with conclusions based on a dataset of 15 human prompts** (even if there are artificially inflated). Especially since the main task is classification into 2 classes.

**Key Questions For Authors:**

The whole paper conclusions are based on 15 unique human prompts [W1], this is unacceptable. The method is over-complicated [W3] and it does not solve one of the key aspects of the problem [W4] while it does not compare against a simple baseline approach [W6].

**Limitations:**

Yes.

**Strengths And Weaknesses:**

### **[W1 (soundness)]** Dataset construction
The dataset contains 5000 instances but those instances are only made from 228 unique human prompts. All dataset instances are automatically generated and labeled using GPT-4o without human verification. The authors manually annotate 100 dataset instances and report 90% agreement but they don’t specify how many humans participated in this evaluation, how many humans judged each instance, and what is the human agreement rate. Evaluation on tiny correlated dataset
The authors evaluate only on 200 dataset instances coming only from 15 unique human prompts. **The conclusions of this paper are based on 15 unique human prompts!** This is preposterously small number. For proper analysis and comparison with related works, different dataset must be used. The contribution of this work cannot be properly judged given this major issue. Also, it is not clear what the train set is used for.

### **[W2 (presentation)]** Missing definitions and unclear wording
In section 4.1:
- ’t' is both step and position. Neither is explained.
- It is unclear what M — “token-to-segment mapping optimized for balanced allocation” — is.
- Segment value E is undefined.
- Symbols V, y, k, v, l, h don’t even have names. They are missing definitions completely.

What are the terms: intent binding (payload encoding), entropy-gated embedding (selective injection), and confidence-based decoding (erasure-aware recovery)? Why is there always a different term in brackets? Since even the authors thought that the terms are not self-explanatory and they need explanation in the brackets, why to introduce them at all and why not to use just the terms in brackets?

### **[W3 (significance)]** ECC
Why do the authors use ECC to embed two messages? It is clear that having two messages of length k: one of all zeros and one of all ones is has the maximum distance between the two messages and allows for up to k/2 bit flips. Based on this, it is unclear what the ECC ablation is measuring. There is no ECC scheme that can correct for more than k/2 bits.

### **[W4 (significance)]** Intent classifier
The authors give only a vague options how to construct the intent classifier. However, the method can be only as good as the intent classifier. The authors report only 90% agreement with the dataset labels while removing ambiguous cases. Including the ambiguous cases, this discrepancy may be much larger. Therefore, it can be expected that the intent classifier will decrease substantially the end-to-end accuracy on the task.

### **[W5 (originality)]** The originality of Intent-Adaptive Entropy Gating
The gating to watermark only high entropy tokens has been done previously in works such as “Who Wrote this Code? Watermarking for Code Generation”. However, in the text, it is presented and one of the main contributions in Section 4.


### **[W6 (significance)]** Binary decision instead of p-value
The method retrieves a binary message that is then summed and thresholded to get binary decision (Generated or Edited with a potential third option “unknown”). A big negative of the method is that it does not provide a p-value similarly to, e.g. KGW watermarking. In general the whole scheme is very complicated with unclear benefits, e.g. ECC, multi-bit watermarking when having only 2 classes, etc. Why the authors not use, e.g., KGW watermarking with two different keys — one for the Generated and one for the Edited class. Then they can check for these two keys and, after simple correction for doing the test for 2 keys, can obtain p-value associated with each class. The authors evaluate if their method outperforms this simple setup.

The authors claim that “zero-bit methods like KGW cannot embed intent payloads and thus cannot perform attribution”. It is not true, two different keys can be used to perform the same type of “attribution” done in this paper.

---

> ### Author Rebuttal · Authors · 2026-03-31
>
> We thank the reviewer for the detailed critique. We make two upfront corrections: (a) our claim that "zero-bit methods like KGW cannot embed intent payloads" was imprecise, two different keys can achieve binary attribution, and we now evaluate against this baseline; (b) IACW's contribution is robust binding and recovery of a known intent signal, not post-hoc intent inference (W4).
>
> **W1.** Our benchmark is not "15 prompts": it contains **228 human-written seed prompts** and **5,000 DAJ-expanded instances**; 15 is only original held-out test seed count (each seed expands into multiple instances). The train split is used to train the intent classifier; the watermarking mechanism itself requires no training. We expanded the held-out evaluation(new 100 seeds, 400 instances):
>
> | Setting      | Original | Expanded |
> | ------------ | -------- | -------- |
> | Clean Acc.   | 99.5     | 95.0     |
> | Del-20% Acc. | 95.0     | 89.5     |
>
> The drop stems from shorter outputs in the broader seed pool, which reduce embeddable tokens. Human verification (2 annotators, κ=0.87, 91.5% agreement with Judge; Appendix A) confirms label quality. To our knowledge, no existing dataset provides intent-level annotations for watermark evaluation; we will release IACW-Instruct to facilitate future comparisons.
>
> **W2.** §4.1 builds on the multi-bit backbone of Qu et al. (2025), and we inherited several symbols without re-defining them in our paper. In the revision we will make the notation self-contained: (1) explicitly define `t` as the generation step at first use; (2) formally specify `M` as the token-to-segment mapping `p = M[y_{t-1}]` and the encoded segment value `E[p]`, with consistent notation between equations and Algorithm 1; (3) add definitions for all symbols currently lacking them (`V`, `ℓ_t`, `h`, etc.); (4) rename the secret key to `κ` to resolve its overload with the ECC message dimension `k`; (5) remove the redundant dual terminology (bracketed alternatives) and use a single set of terms throughout.
>
> **W3.** For binary labels under pure bit-flip corruption, majority vote over all-zeros/all-ones achieves k/2 correction and is a legitimate baseline. ECC's value lies in three aspects: (1) erasure handling, when entropy gating skips positions (g_t=0), corresponding segments produce near-uniform votes; majority vote treats these as errors, while our decoder marks them as erasures and BCH corrects 2t erasures vs t errors; (2) low-budget robustness, editing tasks yield far fewer embeddable positions than generation, so ECC is critical in this scarce-budget regime; (3) multi-class scaling, codebook decoding extends naturally to n>2 classes, as demonstrated by our 4-class proof-of-concept. The ablation in Table 6 confirms this: removing ECC drops accuracy from 99.5% to 90.0%.
>
> **W4.** Intent assignment and watermarking are separable. In vertical applications, intent is available from application context, grammar/polishing tools (e.g., Grammarly) are inherently editing-oriented, creative-writing or story-generation platforms (e.g., NovelAI) are generation-oriented, so intent is near-deterministic. In general-purpose chat settings, intent must be inferred from the prompt and upstream errors can occur. Our contribution is the downstream layer: robust binding/recovery of a provided intent signal.
>
> **W5.** Our contribution is not entropy gating itself but: (1) intent-adaptive dual thresholds, prior methods apply a single fixed threshold regardless of task, whereas we use different thresholds for editing vs. generation to match their different fidelity requirements (τ_edit=0.5 vs τ_gen=0.3); and (2) integration with erasure-aware ECC, where skipped positions are modeled as structured erasures that the code is designed to correct, rather than simply being dropped.
>
> **W6.** We implemented KGW-Dual-Key (k_edit, k_gen). Intent = higher z-score (Unknown if both below threshold). Results across z-thresholds:
>
> | Method       | z-threshold | Clean Acc. | False Attr. Rate |
> | ------------ | ----------- | ---------- | ---------------- |
> | KGW-Dual-Key | 3.0         | 88.0       | 0                |
> |              | 2.0         | 93.5       | 5.0              |
> | **Ours**     | —           | **99.5**   | **0**            |
>
> KGW-Dual-Key faces an inherent accuracy–false attribution tradeoff: lowering the z-threshold improves coverage but introduces false attributions,both keys may exceed the threshold, and higher z-score may correspond to the wrong intent. The gap stems from two structural differences: (1) **uniform embedding**, KGW applies the same bias strength regardless of intent, but editing tasks require conservative embedding to preserve source fidelity, leaving less signal for the z-test; and (2) **text-level vs. segment-level uncertainty**, KGW aggregates evidence into a single whole-text z-score, whereas IACW accumulates segment-level signals through erasure-aware ECC decoding, recovering intent even when individual segments are weak.

---

> > ### Author Rebuttal · Reviewer_LcbK · 2026-04-03
> >
> > I thank the authors for the KGW dual key experiment. However, I should point out that this and other experiments such as Gumbel-max with two keys should have been part of the submission from the start. The presented watermarking is also not distortion free, unlike Gumbel-max, for example.
> >
> > Unfortunately, the author do not address **[W1]** that their evaluation is based only on 15 human prompts. **I simply cannot recommend to accept a paper to ICML with conclusions based on a dataset of 15 human prompts** (even if there are artificially inflated).

---

> > > ### Author Response · Authors · 2026-04-07
> > >
> > > We thank the reviewer for acknowledging our KGW-Dual-Key experiment and for the continued engagement. We would like to offer two further clarifications.
> > >
> > >  **1. Dataset scale (W1)**
> > >
> > >  We would like to clarify the distinction between corpus scale and the original held-out split. IACW-Instruct is built from **228 human-written seed prompts**; the 15-seed number refers only to the original held-out test split in the submission. To directly address the concern that this seed-level evaluation may be too narrow, we additionally ran an **expanded evaluation on 100 new seeds (400 instances)**, obtaining Clean Acc. 95.0% and Del-20% Acc. 89.5%.
> > >
> > >  Regarding data construction: our approach, using LLMs to diversify human seed prompts across scenarios, follows the same methodology behind widely adopted datasets such as Self-Instruct (Wang et al., 2023), Alpaca (Taori et al., 2023), and UltraChat (Ding et al., 2023), as well as prompt optimization frameworks such as DSPy (Khattab et al., 2024) and EvoPrompt (Guo et al., 2024). For reference, Alpaca uses 175 human-written seeds and expands them to 52K instances via GPT-3.5, a seed-to-instance ratio far larger than ours (228 → 5,000). To our knowledge, no existing dataset provides intent-level annotations for watermark evaluation; we will release IACW-Instruct to support future comparisons.
> > >
> > >  **2. Gumbel-max with two keys (W6)**
> > >
> > >  Thank you for raising the Gumbel-max comparison. We agree that its distortion-free property is an advantage for text quality. However, the core limitation we demonstrated with KGW-Dual-Key is inherent to the **multi-key detection paradigm itself**, not specific to KGW, where Gumbel-max dual-key would face the same structural challenges:
> > >
> > > 1. **O(N) detection complexity**: N intents require N independent statistical tests, regardless of the underlying zero-bit method.
> > >
> > > 2. **False attribution**: As N grows, the probability that any key's test statistic exceeds the threshold by chance increases. Our KGW-Dual-Key results confirm this empirically: lowering the z-threshold from 3.0 to 2.0 raises false attribution from 0% to 5.0%.
> > >
> > > 3. **Theoretical grounding**: As Qu et al. (2025) discuss, naive multi-key extensions assign an independent key per message value, but mismatched keys behave as independent noise, resulting in poor separation under constrained token budgets.
> > >
> > >  The distortion-free property improves generation quality but does not resolve the detection-side tradeoff between coverage and false attribution. In essence, dual-key detection replaces multi-bit decoding with zero-bit testing plus key enumeration, precisely the scalability challenge that motivated the development of multi-bit watermarking methods. Our KGW-Dual-Key experiment empirically validates this theoretical expectation.
> > >
> > > We hope these clarifications address the remaining concerns and are happy to discuss further.

---

### Official Review · Reviewer_2bK5 · 2026-03-13

**Soundness:** 3
**Presentation:** 2
**Significance:** 2
**Originality:** 2
**Overall Recommendation:** 4
**Confidence:** 2

**Summary:**

The paper proposes Intent-Aware Controllable Watermarking (IACW), a multi-bit watermarking framework for attributing AI intent in LLM-assisted writing, specifically distinguishing Editing from Generation. The method combines intent-adaptive entropy gating with confidence-based erasure decoding over ECC, and encodes the two intents as maximally separated binary codewords. The paper also introduces IACW-Instruct, a 5,000-example benchmark for intent attribution.

**Compliance With Llm Reviewing Policy:**

Affirmed.

**Final Justification:**

I understand the authors’ explanations, although some aspects may still benefit from further investigation in future work (e.g., broader benchmark settings in W2, and clarification of evaluation protocols in Q1). But the paper still has contributions to the new area. Overall, I would raise my score accordingly.

**Key Questions For Authors:**

1. In the ablation study, I am confused by the result that w/o Erasure Decoding achieves a much lower PPL than IACW (Full) (20.8 vs. 42.1). Could the authors clarify the intuition behind this result?

**Limitations:**

yes

**Strengths And Weaknesses:**

Strengths:
1. The paper tackles a meaningful and timely problem. The shift from binary AI detection to provenance of AI participation is well motivated.
2. The paper includes a benchmark contribution in addition to the method. The dataset schema and DAJ construction pipeline are described clearly enough in the main paper.
3. The high-level pipeline (binding → gating → decoding) is clearly laid out, and the ablation in Table 6 is useful.

Weaknesses:
1. The intent-adaptive entropy thresholds are reasonable but remain heuristic. Since the threshold directly controls skipped positions and therefore downstream erasures and abstentions, it would strengthen the paper to include a sensitivity analysis showing how threshold choices affect unreliable segments, Unknown predictions, and overall attribution accuracy.
2. The benchmark appears somewhat “cleaned” relative to real-world writing workflows. Ambiguous mixed-intent cases are explicitly filtered out, and compound requests are later reduced to a dominant label, which makes the task easier but may under-represent the messy intent boundaries that arise in practice. Relatedly, Section 4.2 abstracts intent assignment into an external IntentSource, but the paper does not analyze what happens when that upstream intent label is itself wrong at deployment time; in that case, the system may faithfully embed and recover an incorrect intent.
3. The method may be sensitive to the degree of editing intensity, i.e., the relative contributions of human-authored content and AI rewriting. Maybe it is better to systematically analyze how attribution reliability changes across light polishing versus substantial revision.

---

> ### Author Rebuttal · Authors · 2026-03-31
>
> We thank the reviewer for the constructive and specific feedback.
>
> **W1 (Threshold sensitivity).** Since editing requires stricter fidelity preservation (τ_edit ≥ τ_gen by design, §4.3), we sweep over the valid region τ_edit ≥ τ_gen:
>
> | τ_edit \ τ_gen | 0.3      | 0.5  | 1.0  |
> | -------------- | -------- | ---- | ---- |
> | 0.5            | **99.5** | 98.0 | —    |
> | 1.0            | 97.0     | 97.0 | 95.5 |
>
> All configurations achieve ≥95.5% accuracy, confirming the method is robust to threshold choice. All configurations remain within 4pp of the optimum.
>
> **W2 (Benchmark design).** The benchmark is intentionally controlled relative to open-ended real-world writing. We filtered mixed/compound-intent cases to isolate the core question of this paper: *given an intent signal, can a watermark reliably bind that intent to text and later recover it under attacks?* We will clarify this scope explicitly in the revision.
>
> The source of intent depends on deployment. In vertical/specialized applications, intent is often available from application context, grammar/polishing tools (e.g., Grammarly) are inherently editing-oriented, creative-writing or story-generation platforms (e.g., NovelAI) are generation-oriented, so intent can be supplied directly by the application layer. In general-purpose chat assistants, however, editing vs. generation must often be inferred from the prompt, and upstream intent errors can occur.
>
> This does not invalidate the watermarking mechanism: intent assignment and intent-preserving watermarking are separable components. If the upstream intent source is wrong, the system will bind the provided label, so end-to-end performance depends on both upstream intent accuracy and downstream recovery robustness. Our contribution is the latter conditional layer, which is needed regardless of how the intent is obtained. We acknowledge that robust intent inference in open-domain chat is outside this paper's scope. We will revise the paper to make this distinction explicit.
>
> **W3 (Editing intensity).** To isolate the effect of editing intensity, we generated 100 instances per sub-type and evaluated independently:
>
> | Sub-type  | Intensity | Clean Acc. (n=100) |
> | --------- | --------- | ------------------ |
> | Grammar   | Light     | 78                 |
> | Polish    | Medium    | 89                 |
> | Summarize | Heavy     | 91                 |
> | Rewrite   | Heavy     | 92                 |
>
> The trend confirms that attribution reliability scales with editing intensity: light edits produce very few LLM-generated tokens, leaving minimal embedding capacity, this is an expected scarce-budget regime rather than a method limitation. For medium and heavy edits, entropy gating finds sufficient high-entropy positions to embed a robust signal. Summarize, despite being heavy in text transformation, has slightly lower accuracy than Rewrite because summarization compresses the source, yielding fewer total watermark-carrying tokens.
>
> We acknowledge that light editing represents a genuine scarce-budget limitation; extending IACW to handle minimal-edit scenarios (e.g., via cross-output signal aggregation) is a concrete direction for future work.
>
> **Q1 (PPL: w/o Erasure Decoding 20.8 vs Full IACW 42.1).** The PPL values in Table 6 are computed on successfully attributed (non-Unknown) samples, which differ across decoder configurations. Since w/o Erasure Decoding produces more Unknown predictions (lower coverage), its PPL is computed on a smaller, easier subset. This is a reporting artifact, not a quality difference. We will report PPL on the full sample set in the revision.

---

> > ### Author Rebuttal · Reviewer_2bK5 · 2026-04-03
> >
> > I understand the authors’ explanations, although some aspects may still benefit from further investigation in future work (e.g., broader benchmark settings in W2, and clarification of evaluation protocols in Q1). But the paper still has contributions to the new area. Overall, I would raise my score accordingly.

---

### Official Review · Reviewer_KQDx · 2026-03-14

**Soundness:** 3
**Presentation:** 3
**Significance:** 4
**Originality:** 4
**Overall Recommendation:** 5
**Confidence:** 4

**Summary:**

The paper has three contributions:
* IACW-Instruct - 5000 samples benchmark that contains diverse edits and generation tasks constructed using GPT-4o with Director-Actor-Judge pipeline starting from a seed of 228 real user prompts.
* Entropy-Gated Embedding - selective injection of watermark bits only at positions where the token distribution is high, with indent dependent thresholds (lower for editing and higher for generation)
* Confidence-Based Erasure Decoding - a method that identifies unreliable segments at the inference time using confidence voting and treats them as reassures for ECC recovery. The indent is encoded and a binary payload using codewords (edit - all zeros, generate - all ones) with BCH error correction codes. Results on Llama-3-8B-Instruct demonstrate 99.5% intent detection accuracy on clean text, 95% accuracy under 20% token deletion, near-underwatermarked semantic fidelity on editing tasks and 100% watermarking detection accuracy. The paper also evaluates robustness of paraphrasing and back-translation and mixed human written and AI generated documents.

**Compliance With Llm Reviewing Policy:**

Affirmed.

**Key Questions For Authors:**

1. Any evaluation on models other than Llama-3-8B-Instruct? Even preliminary numbers on Mistral-7B or a larger Llama would help. Without this, the generalization claim is unsupported.

2. How would the extension to a n-class system (ex. polish, rewrite, summarize, generate) look like?

**Limitations:**

Yes

**Strengths And Weaknesses:**

Strengths:
* The problem framing is the best part of the paper. Moving from “AI or not?” to “how was AI used?” can be potentially useful governance. Even if the two class intent is limited, the intent attribution as a watermarking objective is valuable.
* IACW-Instruct is unique as the reviewer is not aware of any prior datasets that have intent annotations for watermarking studies.

Weaknesses:
* The binary edit/generate taxonomy is the biggest problem. It's too simplistic. Their own data shows 41% of dataset rejections were due to intent ambiguity - that suggests the boundary is inherently fuzzy. Many real tasks are mixed (ex. rewrite and add a summary). The reviewer is curious how the all-zeros/all-ones codeword design extends to more categories without fundamentally redesigning the ECC setup.
* Single-model evaluation is the main weakness of this work. The reviewer is willing to change the review if more model results are supported.

---

> ### Author Rebuttal · Authors · 2026-03-31
>
> We sincerely thank the reviewer for the positive and high-confidence evaluation.
>
> **Q1 (Multi-model) — also addressing the single-model weakness.** We extended to Mistral-7B and Qwen2.5-7B-Instruct:
>
> | Model      | Clean Acc. | Del-20% | Paraph. |
> | ---------- | ---------- | ------- | ------- |
> | Llama-3-8B | 99.5       | 95.0    | 69.0    |
> | Qwen2.5-7B | 99.0       | 96.5    | 74.0    |
> | Mistral-7B | 93.0       | 88.5    | 73.0    |
>
> IACW operates on token probability distributions and is model-agnostic by design; the framework requires no model-specific adaptation.
>
> **Q2 (N-class extension) — also addressing the binary taxonomy concern.** The binary case maps Edit and Gen to maximally separated messages (0^k and 1^k), as described in §4.2. To extend to N intents, we select N messages whose BCH-encoded codewords have large pairwise separation. The encoding, extraction, and erasure-handling pipeline all remain unchanged; only the codebook changes. For illustration, with the paper's (15, 7) BCH code (k=7 message bits, n=15 codeword length, 2-bit error correction):
>
> | Intent    | Message (k=7 bits) |
> | --------- | ------------------ |
> | Polish    | 0000000            |
> | Rewrite   | 0011101            |
> | Summarize | 1100110            |
> | Generate  | 1111111            |
>
> Fitting 4 codewords in the same code space reduces pairwise distance compared to the binary case, lowering error tolerance. Our 4-class proof-of-concept:
>
> | Setting                   | Clean Acc. |
> | ------------------------- | ---------- |
> | Binary (Edit/Gen)         | 99.5%      |
> | 4-class (Pol/Rew/Sum/Gen) | 92.0%      |
>
> This granularity-robustness tradeoff is expected: finer categories reduce inter-class codeword distance and therefore attack tolerance (92.0% vs. 99.5%). The high ambiguity/rejection rate (41%) in our annotation study further indicates that fine-grained intent taxonomies are harder to annotate consistently, increasing label noise, this motivates binary as the default even though N-class is mechanically supported. Binary edit-vs-generate is our deployment default because this boundary is empirically stable and aligned with the governance question we target ("did the user provide the ideas?"). For mixed-intent prompts (e.g., rewrite plus summarize), a single-label taxonomy is inherently limited; we will add a discussion of hierarchical and compositional intent as future directions. To support larger N while maintaining robustness, one can increase the codeword length n (e.g., from (15, 7) to (31, k')) or choose a higher-distance BCH code; the pipeline architecture remains the same.

---

> > ### Author Rebuttal · Reviewer_KQDx · 2026-04-03
> >
> > Concerns have been fully answered, and will keep the original positive score as is. Thank you!

---

### Official Review · Reviewer_iZds · 2026-03-16

**Soundness:** 3
**Presentation:** 3
**Significance:** 3
**Originality:** 3
**Overall Recommendation:** 4
**Confidence:** 2

**Summary:**

The paper proposes shifting AI governance from binary detection ("was AI used?") to intent attribution - distinguishing whether AI edited existing content or generated content from scratch, embedded proactively via watermarking. The authors claimed contributions are:

1. Formalization of Intent Attribution.
2. IACW-Instruct -5,000-instance benchamrk with structured intent annotations across 9 editing and 10 generation sub-types, built via a Director–Actor–Judge pipeline.
3. IACW framework - achieving 99.5% attribution accuracy on clean text, 95% under 20% token deletion, and practical utility in mixed human-AI documents down to 30% AI-polished content.

**Compliance With Llm Reviewing Policy:**

Affirmed.

**Key Questions For Authors:**

1. Could you clarify what instructions, participants were given when contributing seed prompts. For example, were they asked to generate prompts freely, assigned specific task scenarios, or were prompts collected from their organic LLM usage?

2. You might benefit from more clearly justifying why this particular binary Edit/Gen taxonomy is the appropriate level of granularity for real governance or policy decisions, and explicitly explaining why you chose it over plausible finer- or coarser-grained alternatives.

**Limitations:**

Yes.

**Strengths And Weaknesses:**

Strengths:
1. IACW-Instruct provides 5,000 instances with structured intent annotations across 9 editing and 10 generation sub-types, built on real user seeds with IRB approval and careful seed-level splits that prevent near-duplicate leakage across partitions.
2.Reported numbers are compelling for the claimed setting: near-unwatermarked quality on editing tasks, 99.5% clean intent attribution accuracy, and 95% accuracy under 20% token deletion, which substantiate the central claims about fidelity–robustness trade-offs.
3.The paper cleanly formulates “intent attribution” as a watermarking problem and ties it to concrete governance and fairness concerns, which gives the work a clear motivation.


Weaknesses:
1. All experiments use a single model (Llama-3-8B-Instruct), which weaken generalizability across LLM families.
2. The paper does not include a prompt-free post-hoc intent classifier baseline, making it impossible to assess how much of the attribution performance is attributable to the watermark signal versus stylistic differences between edited and generated text that any classifier could exploit.
3. While IACW-Instruct is carefully constructed, the dataset contribution feels somewhat under-positioned, as the paper does not explicitly compare it to existing co-writing, editing, or provenance datasets to show what new evaluations it uniquely enables.

---

> ### Author Rebuttal · Authors · 2026-03-31
>
> We thank the reviewer for the positive assessment and thoughtful suggestions.
>
> **W1.** We extended to Mistral-7B and Qwen2.5-7B-Instruct:
>
> | Model      | Clean Acc. | Del-20% | Paraph. |
> | ---------- | ---------- | ------- | ------- |
> | Llama-3-8B | 99.5       | 95.0    | 69.0    |
> | Qwen2.5-7B | 99.0       | 96.5    | 74.0    |
> | Mistral-7B | 93.0       | 88.5    | 73.0    |
>
> IACW operates on token probability distributions and is model-agnostic by design. The framework itself requires no model-specific adaptation.
>
> **W2.** We fine-tuned a DeBERTa-v3-large classifier on the IACW-Instruct training set (unwatermarked outputs only).
>
> | Method              | Clean Acc. |
> | ------------------- | ---------- |
> | Post-hoc Classifier | 55.0       |
> | IACW (Ours)         | 99.5       |
>
> The 55.0% accuracy (near-random) confirms that stylistic differences between edited and generated outputs are insufficient for intent attribution, the attribution performance is almost entirely attributable to the watermark signal, not exploitable surface features.
>
> **W3.** We compare IACW-Instruct with three adjacent resources on dataset-level features:
>
> | Dataset           | Intent label      | Paired user draft  | Perturbation suite |
> | ----------------- | ----------------- | ------------------ | ------------------ |
> | Mysore (2025)     | ✗ (goal-oriented) | ✗                  | ✗                  |
> | Trove (2025)      | ✗                 | ✗ (source ≠ draft) | ✗                  |
> | **IACW-Instruct** | **✓**             | **✓**              | **✓**              |
>
> No existing dataset is constructed to evaluate proactive intent injection into watermarks. Existing resources (including WildChat) are designed for post-hoc interaction analysis or source tracing, not for actively embedding and recovering intent signals. IACW-Instruct fills this gap by providing two dataset-level features that adjacent resources lack: (i) **provenance-oriented intent labels** (Edit/Gen + sub-types) grounded in user-provided anchor presence (Def. 2.1), Mysore annotates goal/genre-oriented writing behaviors but does not systematically distinguish whether the user supplied source content to be modified; and (ii) **paired user drafts and edited outputs** for watermark fidelity evaluation, Trove provides source sentences for retrieval/tracing but not user-provided editing anchors. Beyond dataset fields, IACW-Instruct is also *designed for* y-only attribution (no prompt/history at evaluation time) and adversarial robustness evaluation (paraphrasing, back-translation, token deletion with ground-truth intent labels).
>
> **Q1.** 47 participants from diverse academic backgrounds were instructed to contribute prompts drawn from their real, naturally occurring LLM interactions (e.g., copied from past chat histories), rather than generating new prompts specifically for the study. No predefined task scenarios were assigned, nor were participants asked to invent prompts for hypothetical tasks. Participants could omit any prompts they did not wish to share. All submitted prompts were anonymized to remove PII.
>
> **Q2.** We use Edit/Generation as the default taxonomy because it captures a coarse-grained intent distinction that is both operationalizable and useful for watermarking: whether the output is constrained by user-supplied source content that serves as a semantic anchor (Editing), or produced without such source-level anchors (Generation). We operationalize this distinction via anchor presence/absence (Def. 2.1), which is easier to annotate consistently in our setup than finer-grained labels such as proofreading vs. rewriting. From the decoding perspective, binary intents are a robust operating point under a fixed watermark budget: fewer classes permit larger code distance and therefore lower confusion (Sec. 4.2). In practice, fine-grained subtypes such as polish vs. rewrite are difficult to annotate reliably, our dataset construction saw 41% of rejections due to intent ambiguity at the sub-type level, while the edit-vs-generate boundary is substantially more stable. We do not claim binary is the only useful taxonomy; rather, it is a clear and robust default for our setting. Our 4-class proof-of-concept further shows that the framework extends to finer granularity when needed:
>
> | Setting                   | Clean Acc. |
> | ------------------------- | ---------- |
> | Binary (Edit/Gen)         | 99.5       |
> | 4-class (Pol/Rew/Sum/Gen) | 92.0       |
>
> The 7.5pp drop illustrates the expected tradeoff: finer categories reduce inter-class Hamming distance and increase boundary ambiguity, confirming that binary is a robust default for the most governance-salient boundary.

---

> > ### Author Rebuttal · Reviewer_iZds · 2026-04-08
> >
> > My original concerns have been adequately addressed by the authors. Taking into account the other reviewers' comments, I have decided to keep my current score, which is already at or above the acceptance threshold.

---

### Decision · Program_Chairs · 2026-04-30

**Decision:**

Accept (regular)

**Comment:**

The paper has received three reviews in favor of acceptance (2x WA, 1x A), and one strongly critical review (1x R).

Reviewer KQDx is supportive of the novel problem framing fusing watermarking and intent, and maintains their clear accept score post-rebuttal, noting that their outstanding queries are addressed (notably exploration of other models).

Reviewer iZds also raises the model generality point, and queries whether experiments are cofounded by stylistic bias.  Whilst they do not raise their score from weak accept (WA) they do note their concerns are fully addressed by the rebuttal.

Reviewer 2bK5 has concerns over heuristic choices in the method and the reasonableness of the data considering it may not be representative of real world content.   The rebuttal substantially addresses their concerns, and the reviewer upgrades the score noting also the novelty of the problem setting.

Reviewer LcbK raises a number of issues with the technical method and its analysis, including potential design alternatives (KGW watermarking), concerns with unnecessary ECC ablation and noting certain aspects of the method could better cite lineage to related work.  The main concern however is in the experimental validation, where LcbK is concerned that the test set upon which validation is performed is sourced by augmenting only a small set (15) of human prompts.     In the rebuttal, several of these points are addressed notably experiments using KGW and the provision of core results over a larger test dataset.   The reviewer maintains their score of firm  reject (R) and score on these points suggesting insufficiency of evaluation and further explorations of the KGW suggestion.

The AC considers the paper to make a useful and novel contribution to the watermarking field i.e. shifting from AIGC use detection to attribution of how AI is used i.e. intent.  This framing is novel to watermarking and multiple reviewers explicitly state the value this.  The rebuttal has shown the approach to generalise well across models addressing two reviewers’ major concern.  Reviewer LcbK raises a valuable question on whether the experimental data support the value claimed in this novel avenue of exploration.  The AC agrees that the initial evaluation is weaker, but there is further data in the rebuttal showing broader test set coverage that the reviewer has not responded on.   The AC’s initial view is that on the balance there is sufficient support to accept the paper, but will table this for further discussion with the SAC.